# Adaptive Vision Transformer for Event-Based Human Pose Estimation

## ABSTRACT

Human pose estimation has made progress based on deep learning. However, it still faces challenges when encountering exposure, low light, and high-speed scenarios such as motion blur and miss human contours in low light scenes. Moreover, due to the extensive operations required for large-scale convolutional neural network (CNN) inference, marker-free human pose estimation based on standard frame-based cameras is still slow and power consuming for real-time feedback interaction. Event-based cameras quickly output asynchronous sparse moving-edge information, which is low latency and low power consumption for real-time interaction with human pose estimators. For further study. this paper proposed a high-frame rate labeled event-based human pose estimation dataset named Event Multi Movement HPE (EventMM HPE). It consists of records from synchronized event camera, high frame rate camera and Vicon motion capture system, with each sequence recording multiple action combinations and high frame rate (240Hz) annotations. This paper also proposed an event-based human pose estimation model, which utilizes adaptive patches to efficiently achieves good performance for the sparse and reduced input data from DVS. The source code, dataset, and pre-trained models will be released upon acceptance.

## CCS CONCEPTS

• **Human Centered Computing** → **Human Computer Interaction (HCI)**.

## KEYWORDS

Event camera, Artificial intelligence, Silicon retina, Biomimetic vision, Graph learning, Semantic segmentation

## 1 INTRODUCTION

Human Pose Estimation is a computer vision task that involves estimating the positions and orientations of body joints and bones from 2D images or videos, which can be used in a variety of applications, such as virtual reality, human-computer interaction, and motion analysis. Although RGB-based human pose estimation [3, 4, 10, 22] has made notable advancements, these approaches frequently encounter barriers due to sensor constraints when confronted with rapid human movements and challenging lighting conditions. By

*ACM MM, 2024, Melbourne, Australia*
© 2024 Copyright held by the owner/author(s). Publication rights licensed to ACM.
ACM ISBN 978-x-xxxx-xxxx-x/YY/MM
https://doi.org/10.1145/nnnnnnn.nnnnnnn

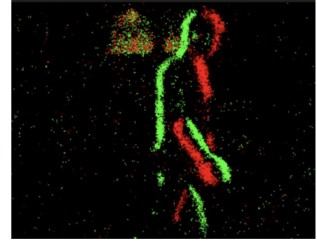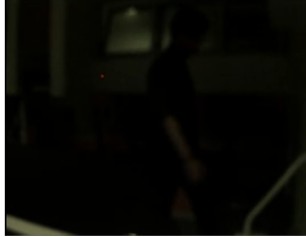

**Figure 1: Comparison of event data and RGB image in low-light scenario. Event cameras provide more information than traditional cameras.**

contrast, event cameras as bio-inspired sensors, offer high temporal resolution, high dynamic range, and low energy consumption. Therefore, event cameras can robustly function in degraded conditions, Figure 1 illustrates an example where event data outperforms RGB image in low-light scenarios. Event cameras have been employed for a multitude of vision tasks, such as optical flow estimation [8, 20] and object detection [14, 26]. Particularly, abundant spatial and temporal information provided by event cameras opens up new possibilities for advancing human pose estimation.

In effectively harnessing the spatiotemporal information inherent in event data, a common approach involves initially converting such data into a frame format, followed by the utilization of Convolutional Neural Networks (CNNs) to capture pertinent features. Nonetheless, CNNs inherently lack the ability to extract temporal cues, constraining the overall performance of human pose estimation. Therefore, some works have introduced Recurrent Neural Networks (RNNs) and Long Short-Term Memory Networks (LSTMs) to enhance the capability in modeling temporal dynamics. However, these methods require significant memory allocation to preserve historical data. Based on this, the Spiking Neural Networks (SNNs) have also been utilized to extract temporal features from event data; however, it has yet to achieve satisfactory performance due to challenges such as training difficulties.

In contrast to the aforementioned methods, the transformer exhibits the capability to model global spatiotemporal characteristics through the utilization of the self-attention mechanism, thereby offering the potential to leverage the advantages of events. For example, Fu et al. [7] introduced transformers to extract target and motion cues for event-based tracking; Gehrig et al. [9] showed that recurrent vision transformers can reach state-of-the-art performance in object detection with event cameras. However, these methods ignore that the computational complexity of the self-attention in transformers is quadratic in the number of patches and tokens, resulting in an unaffordable inference time. This defeats the purpose of using event cameras for high temporal resolution and low latency.

In this paper, we propose an adaptive and efficient visual transformer framework for event-based human pose estimation to tackle the aforementioned issues, based on the unique characteristics of event data. Specifically, our approach mainly contains two key adaptive strategies: (i) adaptive patch sampling: since the event camera records data solely based on lighting changes, the resulting event data is highly sparse, presenting significant activity and inactivity at pixel locations over time. Therefore, the purpose of our adaptive patch sampling scheme is to eliminate inactivity patches by assessing the entropy of the events before inputting to transformers; (ii) adaptive token reduction: this strategy selectively removes less informative tokens in transformers layers utilizing a dynamic token pruning algorithm. This is achieved by evaluating the contribution of each token based on entropy or attention scores, ensuring only tokens with significant informational value are retained. These two mechanisms allow our approach to maintain high performance while reducing computational overhead, making it highly effective for event-based human pose estimation in complex environments.

To exploit event-based visual cues in human pose estimation, we construct a large-scale event-based human pose estimation dataset, named EventMM HPE. It collects 76 human pose estimation sequences consisting of 21 different actions of 7 subjects. The annotation frequency is up to 240Hz. To facilitate future research on multimodal human pose estimation, our dataset provides synchronized event data and RGB images. To the best of our knowledge, EventMM HPE is the dataset with the highest annotation frequency, which is more in line with the original intention of using event cameras.

To sum up, our contributions are as follows:

- We propose a novel adaptive vision transformer architecture for human pose estimation, allowing us to effectively and efficiently extract spatial-temporal features from events.
- We construct a large-scale event-based dataset for human pose estimation. The dataset provides a wide diversity in action and offers high annotation frame rate.
- Extensively experimental results validate that the proposed approach outperforms state-of-the-art methods.

## 2 RELATED WORK

### 2.1 Event-based human pose estimation

Most existing event-based human pose estimation methods directly apply existing traditional framed-based methods [4, 5, 17, 22] by accumulating asynchronous events into frames [2, 13]. However, the sparsity of event data limits the performance of these methods, as event cameras only output events asynchronously in locations with significant brightness changes in the scene. It can be seen that event cameras can only capture moving body parts and ignore some stationary body parts, resulting in incomplete or even missing body parts. Therefore, in long-term sequences, the body will always partially "disappear" in certain frames.

To solve above question, Zhanpeng Shao et al. [18] pproposed using LSTM recurrent networks to achieve geometric consistency and temporal dependence between frames, in order to help recover and complete these lost information. Through a basic cyclic architecture, a newly proposed time dense connection is adopted in a series of time steps to capture the geometric consistency of human

pose between local and distant frames, in order to recover lost human pose information in event frames. Shihao Zou et al. [28] proposed a two-stage deep learning method called EventHPE. This method aims to use two modalities, event and optical flow, to more accurately estimate the three-dimensional human pose. Firstly, in the first stage, FlowNet is used for unsupervised learning to infer the optical flow in the event, where event frame is input into a CNN model to predict optical flow. This optical flow can provide clear geometric information to describe human motion. Then, in the second stage, the output of FlowNet is used as the input of ShapeNet. In this module, the CNN module is used to extract the vectorized feature representation and pass it to the RNN module to infer Internal pose change $\Delta\theta_{t_i}$ in the time interval $(t_{i-1}, t_i)$. In a clear starting posture $\theta_{t_0}$ and shape, then estimate each time point $t_i$ The human posture of $i$. more accurately. This method trains without supervision, thus reducing the need for manual annotation of data. The use of event and optical flow modes to estimate human posture has multiple advantages. Firstly, events and optical flow are closely related to human movement, thus providing more accurate information to describe human posture. Secondly, using optical flow as an important information for estimating human posture can greatly reduce the amount of input data required. This means that grayscale image streams other than events can not be used as input data and only require the use of a single camera. Finally, using ShapeNet to estimate shape changes over time can provide a more accurate estimation of human posture.

The existing methods for human pose estimation based on event cameras mostly simply stack event information into event frames, and then perform related processing such as human pose estimation. The advantage of this method is that data processing is relatively simple, but its disadvantage is also obvious, which is the loss of time information inherent in event information, which cannot fully leverage the advantages of event cameras. Therefore, how to fully utilize event stream data and fully leverage the advantages of event cameras to improve the accuracy of human pose estimation is particularly crucial.

### 2.2 Vision transformer for human pose estimation

Human pose estimation has undergone rapid development from neural CNNs [22] to visual transformation networks. Early work tended to view transformers as better decoders [11, 12, 24], for example, TransPose [24] directly processed CNN extracted features to model global relationships. TokenPose[12] proposes a token based representation by introducing additional tokens to estimate the position of occluded keypoints and modeling the relationships between different keypoints. In order to remove the cellular neural network used for feature extraction, an HRFormer [25] using a transformer to directly extract high-resolution features is proposed. In order to gradually integrate the multi-resolution features in HRFormer, a refined parallel transformer module is proposed. These transformer based attitude estimation methods have achieved superior performance on popular keypoint estimation benchmarks. However, they either require cellular neural networks for feature extraction or require careful design of transformer structures. There has been little effort made in exploring the potential of ordinary

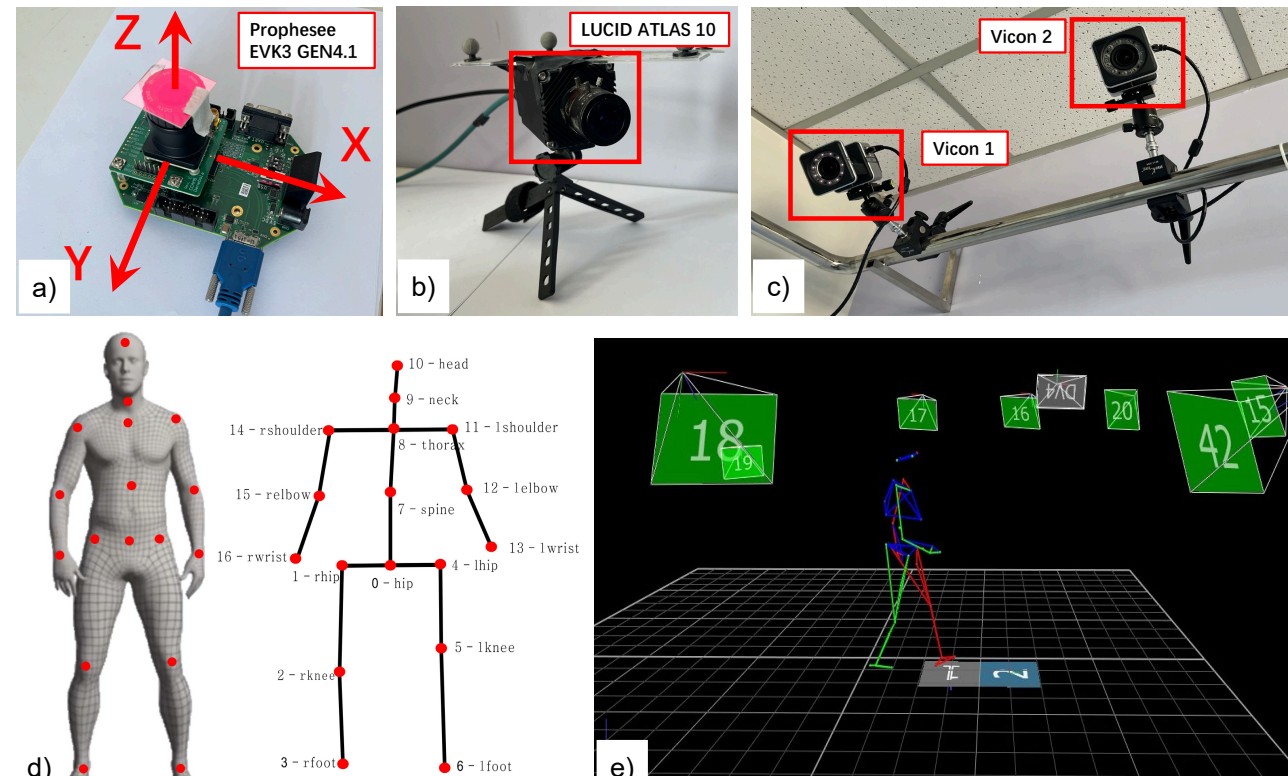

Figure 2: a-c) Prophesee camera, High frame-rate camera and Vicon IR Camera. d) Vicon key positions on the subject and skeleton reputation. e) Schematic of the setup, with Prophesee master camera (position 1), High frame-rate camera (position 2) and Vicon origins. $\varphi_N$.

visual transformers for pose estimation tasks. ViTPose[23] fill this gap by adopting a simple yet effective baseline model. For further efficiency, we propose a adaptive vision transformer based on the plain vision transformers.

## 3 EVENTMM HPE DATASET

Recently, Calabrese et al. [2] constructed the first even-based dataset for 3D human pose estimation by using synchronized 4 event cameras. This work demonstrated that event cameras could enable more efficient human pose estimation towards real-time and power-constrained application. However, the motion sequences in the DHP19 dataset are characterized by simplicity and slowness, thus constraining the pre-trained model's capacity for generalization in real-world environments. The MMHPSD [28] dataset exhibits diversity in human movement speed. However, the accuracy of annotation cannot be guaranteed as it relies on predictions from existing human pose estimation methods. The CDEHP [18] dataset was collected under varying light conditions, but its size remains relatively small. To enable further research on event-based huamn pose estimation, we collect a large-scale dataset termed EventMM HPE. Our dataset collects 76 human pose estimation sequences consisting of 21 different actions of 7 subjects. Each subject is labeled

with 33 joint points. The annotation frequency is up to 240Hz. To encourage community research on multi-modal fusion, we have also gathered synchronous high-frame-rate RGB images with 120FPS. Table 1 provides a comparison of different datasets in terms of number of actions, scenes, resolution, number of frames, mark, and annotation rate.

In summary, compared to other event-based human pose estimation datasets, our dataset offers several advantages: (i) high resolution (1280×720) and high annotation frequency (240Hz), maximizing the utilization of event cameras; (ii) provision of 120FPS RGB images, enabling the fusion of high frame rate multi-modal data; and (iii) diverse action sequences. During data collection, each action sequence captures a wide range of movement speeds and poses, resembling real-life scenarios more closely.

### 3.1 Data Collection and Annotation

**Setup.** As shown in Figure 2 (a)-(c), our EventMM HPE dataset is recorded by the Prophesee EVK3 GEN4.1 event camera, which equips a 1280×720 pixels dynamic vision sensor (DVS). The simultaneous RGB images are recorded by a high-frame-rate camera LUCID ATLAS10 at 120FPS. The ground truth pose location of a human is provided by the VICON motion capture system, which

**Table 1: Table notes: Act represents the type of action, Scenes represents the scene, Resolution represents the camera resolution, Frames represents the number of event frames or the size of data, Mark represents the annotation method used, and Annotation rate represents the frame rate of data annotation.**

| Dataset | Act | Scenes | Resolution | Frames | Mark | Anno. rate |
|---|---|---|---|---|---|---|
| DHP19 | 33 | Indoor | 344*260 | 87k | Vicon (8) | 100 |
| MMHPSD | 12 | Indoor | 1280*800 | 240k | RGB-D | 15 |
| CDEHP | 25 | Outdoor | 1280*800 | 82k | RGB-D | 60 |
| **Ours** | 26 | Indoor | 1280*720 | 1,296k | Vicon (11) | **240** |

captures motion with a high sampling rate (up to 330Hz) and sub-millimeter precision. The VICON motion capture system records the 3D coordinates of the subject's 33 marked joints, identified by markers located on the head, neck, spine, left/right shoulders, chest, left/right elbows, left/right hands, hips, left/right hips, left/right knees, and left/right feet, etc, as shown in Figure 2 (d). Figure 2 (e) illustrates an example of capturing human joint positions using VICON motion capture system. The synchronization method for all three devices is the same as in DHP19 [2].

**Collection and Annotation.** The dataset collection and annotation are divided into three steps: (i) data collection: markers are placed in human body, and the three-dimensional spatial positions of joints are captured under the VICON system, while event data and RGB images are captured using the Prophesee EVK3 GEN4.1 and LUCID ATLAS10, respectively. (ii) event camera intrinsic calibration: the event camera's intrinsic parameters are first calibrated by capturing images of a calibration board using Zhang's calibration method [27]; (iii) data annotation: intrinsic parameters will aid in coordinate system transformation between the event camera and VICON system, facilitating the transformation from 3D points to 2D points.

**Table 2: List of recorded movements**

| S1: Simple Action | S2: Sports Action | S3: Combination Action |
|---|---|---|
| 1. Limp | 1. Play hopscotch | 1. Sweep + mop the floor |
| 2. Use crutches | 2. Lift dumbbells | 2. Kick + walk |
| 3. Climb | 3. Play badminton | 3. Shoot hoops + play ball |
| 4. Goose-step | 4. Hold a toy gun | 4. Walk + wave |
| 5. Sweep the floor | 5. Play table tennis | 5. Squat + stretch |
| 6. Push/pull objects | 6. Box | 6. Run + walk |
| 7. Drink water | 7. Move freely | 7. One-leg jump + jump rope |

## 3.2 Action Analysis

As shown in Table 2, we categorize human body actions into three types: (i) Simple actions, comprising sequences with only basic hand and foot movements; (ii) Sports actions, which involve activities related to sports; and (iii) Combination actions, encompassing sequences with combinations of various poses. Figure 3 provides some visual examples. Figure 3 further demonstrates three sessions

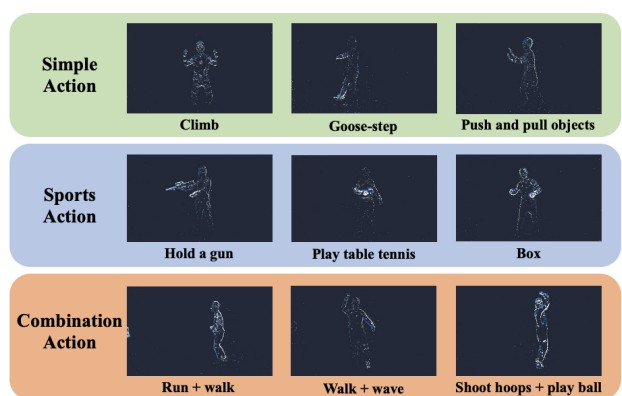

**Figure 3: Movement Samples of different sessions**

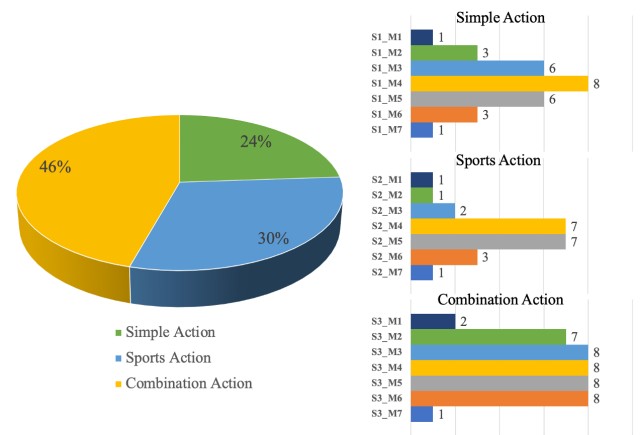

**Figure 4: Data distribution**

of movements including simple session, sports session and combination session, which is more diversity and general. Movements sample of three sessions are as shown in Figure 4. Among them, the combination sessions includes a combination of two consecutive movements. And the three sessions of movement sequences all follow a normal distribution pattern. These statistics indicate the captured EventMM HPE offers wide action and pose diversity.

## 4 PROPOSED METHOD

In this section, we introduce our human pose estimation method based on event cameras. As the original event stream of event camera is asynchronous and discrete, to further improve event utilization, we first process the event streaming data using Locally-Normalised Event Surfaces (LNES)[16], which retains both events' spatial information and temporal information. In section4.2, we introduce our method (Adaptive Transformer) that adaptively adjusts the inference cost of vision transformer (ViT)[6] for handling variable event input length. We validate our method with the experimental results on our EventMM HPE dataset and public DHP19 dataset in Sec.5.

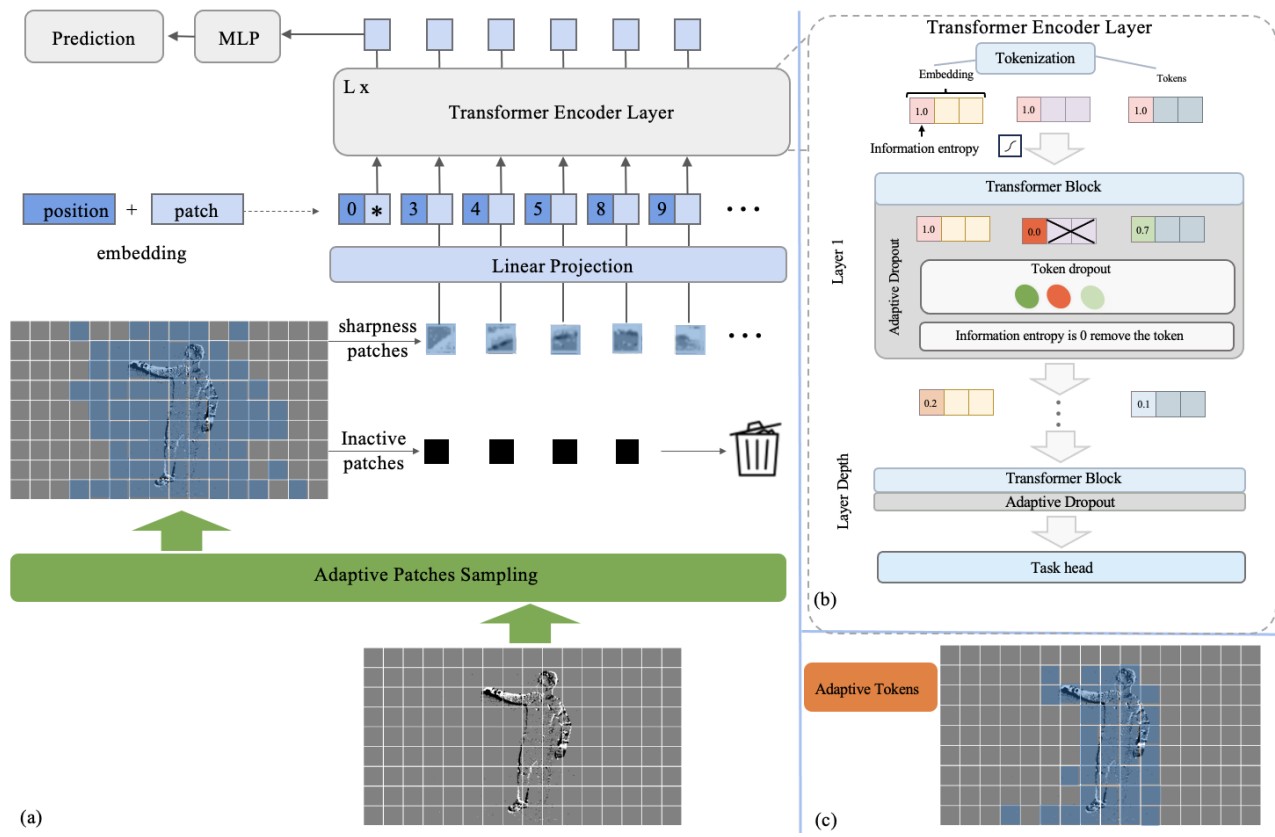

Figure 5: Overview of Adaptive Vision Transformer for event-based human pose estimation, which mainly consist of adaptive patches sampling and adaptive tokens dropout. First adaptive patches sampling module select 85% active patches to retain through information entropy of patches. The remained sharpness patches with more information entropy will be flattened and linearly projected then added with position embeddings. During inference, we propose a information entropy score to adaptive tokens dropout for vision transformers and c) shows the final tokens for using.

## 4.1 Event Processing

Event cameras report local changes of brightness through an asynchronous stream of output events, where consists of spatial coordinate, polarity and temporal timestamp. As temporal information is critical in 3D human pose estimation, we utilize Locally-Normalised Event Surfaces (LNES [16]) to represent event streaming data, retaining both events' spatial information and temporal information. This processing method encodes all events within a fixed time window $L$ as an image $I \in \mathbb{R}^{W \times H \times 2}$ (see Fig. 6, left), which divides them into positive and negative channels based on event polarity, preserving as much information as possible. In contrast to existing representation (*e.g.* [1]), LNES operates with windows-normalised time stamps, specifically as follows:

$$I(x_i, y_i, p_i) = \frac{t_i - t_0}{L} \quad (1)$$

where $x_i$. $y_i$ represents pixel position, $p_i$ represents polarity, $t_i$ represents the time of the current event, $t_0$ represents the starting time of the time window, and $L$ represents the length of the time window.

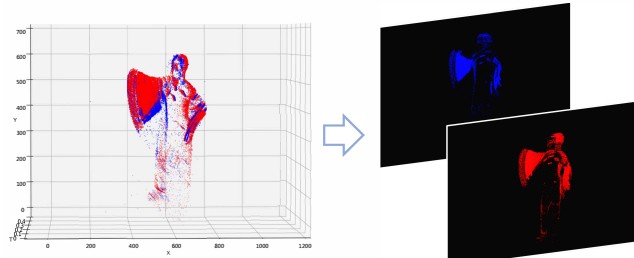

Figure 6: Events representation LNES

## 4.2 Adaptive vision transformer

In this Section, we describe the details of proposed pipeline and motivation. We propose adaptive patches sampling and adaptive tokens dropout based on a vision transformer (ViT) architecture to improve its feature extraction ability and automatically reduce the number of tokens in vision transformers, those are processed in the network as inference proceeds, which is vital for exploiting the

spatial sparsity of event data and the accompanying computation reduction.

### 4.2.1 Adaptive patches sampling.

For a vision transformer network that takes an event image $I \in \mathbb{R}^{C \times H \times W}$ (C, H, and W represent channel, height, and width respectively) as input to make a prediction through:

$$y = C \circ \mathcal{F}^L \circ \mathcal{F}^{L-1} \circ \cdots \circ \mathcal{F}^1 \circ \mathcal{E}(I), \quad (2)$$

where the encoding network $\mathcal{E}(\cdot)$ tokenizes the event image patches from $I$ into the patch embeddings $t \in \mathbb{R}^{N \times E}$, each patch are projected an embedding $E \in \mathbb{R}^{(P^2 \cdot C) \times D}$, patches are divided from $I \in \mathbb{R}^{C \times H \times W}$ with patch size $P^2$, $C$ is the number of patch channels and $D$ is the embedding dimension for each token. $N$ being the total number of tokens, where $N = (H/P) * (W/P)$. $C(\cdot)$ post-processes the transformed class token after the entire stack, while the $L$ intermediate transformer blocks $\mathcal{F}(\cdot)$ transform the input via self-attention.

Not all $N$ patches from LNES event representation contain clear human contours, and patches with low clarity have little information entropy. Some of $N$ patches have low information entropy with no human contour information, which is not valuable for feature extraction in the task and increases unnecessary computational complexity. Our adaptive patches sampling choose 85%$N$ patches to patch embedding, where the patches with more information entropy.

The selected image patches are tokenied an embedding $E$. As shown in Eq. 3 in the first layer of the DVS ViT, the $N$ flattened patches $x_1, \ldots, x_n$ from event image $I$ are embedded as:

$$\mathbf{z}_0 = [\mathbf{x}_{\text{class}}; \mathbf{x}_1 E; \mathbf{x}_2 E; \ldots; \mathbf{x}_n E] + E_{\text{pos}}, \quad (3)$$

where $E_{pos} \in \mathbb{R}^{(N+1) \times D}$ is the positional embedding matrix and $x_{\text{class}} \in \mathbb{R}^D$ is the additional class embedding. Then the embedded vector $z_0 \in \mathbb{R}^{(N+1) \times D}$ is passed into the $L$ transformer encoder layers. To make our adaptive vision transformer more efficiency, we introduce adaptive tokens dropout during the network inference, which means tokens are reducing as our vision transformer block deepens.

### 4.2.2 Adaptive tokens dropout.

Consider the transformer block at layer $l$ that transforms all tokens from layer $l-1$ via:

$$t_{1:n}^l = \mathcal{F}^l(t_{1:n}^{l-1}), \quad (4)$$

where $t_{1:n}^l$ denotes all the $n$ updated token, with $t_{1:n}^0 = \mathcal{E}(x_{1:n})$. Note that the internal calculation process of transformer blocks $\mathcal{F}(\cdot)$ allows the number of tokens $n$ can be changed from a layer to another. This offers out-of-the-box computational gains when tokens are dropped due to low information entropy score. Vision transformer [6, 19] utilizes a consistent feature dimension $E$ for all tokens throughout layers. This makes it easy to learn and capture the global information entropy of all layers in the monitoring joint manager. Compared to CNNs that require clear handling of different structural dimensions at different depths (such as the number of channels), this also makes adaptive tokens dropout easier.

To utilize tokens adaptively, we introduce an input-dependent information entropy score for each token as a using $s_n^l$ for a token $n$ at layer $l$:

$$s_n^l = IE(t_n^l), \quad (5)$$

where $IE(\cdot)$ is a information entropy score. The mechanism of implementing adaptive tokens dropout using changes in information entropy is used in subsequent components in each transformer encoder layer.

The two most important components in each transformer encoder layer are multi-head self attention (MSA) and multi-layer perception (MLP), which are also computationally heavy. The $l$-th transformer encoder layer shown in Fig. 5(b) can be written as:

$$z_l' = MSA(LN(z_l)) + z_l \quad (6)$$

$$z_{l+1} = MLP(LN(z_l')) + z_l', \quad (7)$$

where $l \in \{1, 2, \ldots, L\}$ and $LN(\cdot)$ is the layer normalization function.

The MSA consists of $k$ individual self-attention (SA) heads. Each SA head is formulated as:

$$SA(z_l) = \text{softmax}(q_k^T / \sqrt{D_h}) \cdot v, \quad (8)$$

where

$$[q, k, v] = [z_l U_q, z_l U_k, z_l U_v]. \quad (9)$$

The three learnable matrices $U_q, U_k, U_v \in \mathbb{R}^{D \times D_h}$ projects embedding $z_l \in \mathbb{R}^{(n+1) \times D}$ to $\mathbb{R}^{(N+1) \times D_h}$, where $D_h$ is a dimension of our choice. Thus, each SA head is in $\mathbb{R}^{(N+1) \times D_h}$. The $k$ SA heads operate individually on the embedded patches and then their outputs are concatenated and projected back to $\mathbb{R}^D$ by the trainable projection $U_{MSA} \in \mathbb{R}^{k \cdot D_h \times D}$ in the MSA, formulated as:

$$MSA(z_l) = [SA_1(z_l); SA_2(z_l); \ldots; SA_k(z_l)]U_{MSA}. \quad (10)$$

## 5 EXPERIMENTS

### 5.1 Experiments Setup

**Implementation.** We implement the proposed network in PyTorch. The model is trained using a stochastic gradient descent (SGD) optimizer with a momentum of 0.9 and a weight decay of 5e5. Our network is trained for 20 epochs with batch size 32 on an NVIDIA RTX4090 GPU. The learning rate in [1e-2, 5e-3, 1e-3, 5e-4, 1e-4] decreases as training progresses; for the first 10 epochs, the learning rate is 1e-2; for the remaining 20 epoch, adjusted every 5 epochs.

**Evaluation Metric.** We select MPJPE as our evaluation metric, which means much of the literature reports *mean per joint position error*. For a frame $f$ and a skeleton $S$, MPJPE is computed as

$$E_{MPJPE}(f, S) = \frac{1}{N_S} \sum_{i=1}^{N_S} \left\| m_{f,S}(i) - m_{gt,S}(i) \right\|_2 \quad (11)$$

where $N_S$ is the number of joints in skeleton $S$. For a set of frames the error is the average over the MPJPEs of all frames.

Depending on the evaluation setup, the joint coordinates will be in 3D, and the measurements will be reported in millimeters (mm), or in 2D, where the error will be reported in pixels. For systems that estimate joint angles, we offer the option to automatically convert the angles into positions and compute MPJPE, using direct kinematics on the skeleton of the test subject (the ground truth limb lengths will not be used within the error calculation protocol).

## 5.2 Comparison with State-of-the-art

To validate the effectiveness of our method, we compare the proposed approach with the following three state-of-the-art pose estimation approaches: PointNet[15], DGCNN [21], and DHP19 [2]. The overall pose estimation performance is reported in Table 3 and some samples are shown in Figure 7, which indicate the proposed method offers state-of-the-art pose estimation performance on our proposed EventMM HPE dataset. In particular, our approach outperforms the runner-up by 5.71% and 3.59% in terms of MPJPE on two evaluation subject (*i.e.*, S1 and S2), respectively. We also provide performance on each sequence, which contains a variety of actions and gestures. These results demonstrate the effectiveness of our proposed method. The analysis related to another dataset is provided in the Supplementary Material.

## 5.3 Ablation Study

**Impact of Transformer Parameters.** We choose ViT as the base model and evaluate the effects of different parameters in it on our proposed dataset. Specifically, we exploit the impact of the feature dimension input into the transformer block (*embed_dim*), the number of multi-attention heads (*heads*), the number of transformer blocks (*depth*), the dimension output from the transformer block (*mlp_dim*). As shown in Table 4 and samples shown in Fig, the network performed optimally with an embedding dimension of 512, 6 heads, a depth of 6, and an MLP dimension of 512. Contrary to popular belief, a larger network does not necessarily result in better performance. It appears that the discrepancy lies in the difference

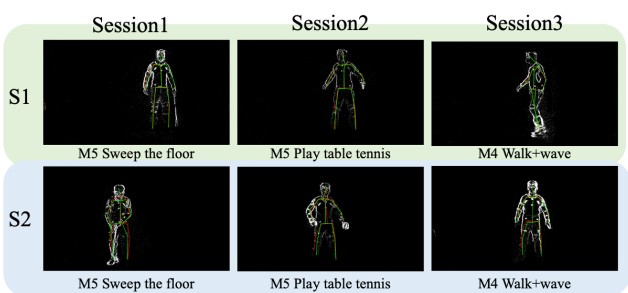

**Figure 7: Event-based human pose estimation Samples on our method**

**Table 4: Ablation results on the ViT-based network.**

| Network params | | | | |
|---|---|---|---|---|
| embed_dim | heads | depth | mlp_dim | MPJPE |
| 1024 | 12 | 2 | 256 | 104.56 |
| 512 | 6 | 6 | 512 | 98.54 |
| 512 | 6 | 12 | 512 | 103.31 |
| 512 | 12 | 6 | 512 | 103.31 |
| 512 | 12 | 12 | 512 | 100.45 |

in data size between our dataset and ImageNet, which boasts a staggering 1.28 million images. However, our dataset remains the event-based pose estimation dataset with the highest frame rate that is currently accessible to the public.

**Impact of Adaptive Patches Sampling.** Our adaptive patches sampling scheme is to eliminate inactive event patches from the input side. To demonstrate its effectiveness, we performed two sets of experiments varying the number of input patches: (i) 400 input patches were downsampled to 340 patches; (ii) 144 input patches were downsampled to 122 patches. The results presented in Table 5 demonstrate that eliminating inactive event blocks leads to an enhancement in prediction accuracy. This is due to the fact that inactive blocks typically lack significant information and may even contain noise that hinders the representation of the network. Since transformer architecture has quadratic memory and time complexity with respect to the number of input patches, thus removing inactive events patches facilitates the deployment of transformers on hardware devices with limited compute resource. For example, by employing our adaptive patches sampling scheme to downsample 400 input patches to 340 patches, the FLOPS of network is diminished from 1.79G to 1.49G. Among two sets, we add random sampling operation with "random" in contrast to our adaptive sampling module, and the results show that with the patches decrease, the performance become better and our adaptive sampling have good performance than random sampling.

**Impact of Adaptive Tokens Dropout Module.** The adaptive tokens dropout module is another key component of our method, as it determines which tokens to discard based on the amount of information they contain during the forward process. To verify the effectiveness of our adaptive tokens dropout module, we conducted ablation experiments on ViT (*i.e.*, head and depth are both 12) and its variants (*i.e.*, head and depth are both 6) respectively with and

**Table 3: The performance of MPJPE on EventMM HPE dataset. S1 and S2 denote two evaluation subject.**

| Seq | | PointNet | DGCNN | DHP19 | **Ours** |
|---|---|---|---|---|---|
| | M4 | 107.75 | 104.95 | 740.84 | 104.34 |
| Session1 | M5 | 106.28 | 103.89 | 732.69 | 93.86 |
| | M3 | 99.5 | 97.95 | 746.32 | 98.53 |
| Session2 | M4 | 97.85 | 94.84 | 725.87 | 100.31 |
| | M5 | 96.41 | 92.58 | 743.36 | 94.7 |
| | M4 | 107.47 | 105.59 | 741.17 | 95.38 |
| | M2 | 107.42 | 105.51 | 726.67 | 104.76 |
| Session3 | M6 | 106.83 | 104.73 | 726.67 | 103.77 |
| | M5 | 108.24 | 105.78 | 728.55 | 100.91 |
| | M3 | 106.23 | 103.93 | 693.10 | 98.87 |
| Mean | | 104.40 | 101.98 | 730.52 | 99.54 |
| Session1 | M4 | 100.45 | 96.42 | 690.28 | 95.38 |
| | M5 | 100.05 | 96.95 | 700.99 | 87.85 |
| | M4 | 107.02 | 104.56 | 751.95 | 93.06 |
| Session2 | M6 | 105.67 | 103.35 | 708.04 | 93.15 |
| | M5 | 95.32 | 91.19 | 723.94 | 89.85 |
| | M4 | 106.04 | 102.94 | 709.56 | 97.74 |
| | M2 | 101.03 | 97.83 | 714.93 | 96.61 |
| Session3 | M6 | 105.19 | 101.96 | 700.42 | 97.27 |
| | M5 | 93.81 | 89.93 | 750.05 | 91.43 |
| | M3 | 106.23 | 93.72 | 742.37 | 88.24 |
| Mean | | 102.08 | 97.89 | 719.25 | 93.06 |

**Table 5: Ablation results of using our adaptive patches sampling scheme.**

| Input Patches | Adaptive | FLOPS | MPJPE |
|---|---|---|---|
| 400 | 400 | 2.99G | 112.55 |
| 400 | 340(random) | 1.84G | 106.01 |
| 400 | 340 | 1.84G | 98.54 |
| 144 | 144 | 1.07G | 104.48 |
| 144 | 122(random) | 892.97M | 103.01 |
| 144 | 122 | 534.21M | 98.25 |

without using our adaptive tokens dropout module. The results in Table 6 show that using this module slightly improves the performance of the base ViT yet significantly reduces the computational complexity. Specifically, when only 32 tokens are retained in the forward process, FLOPs is reduced by 7.47G and the MPJPE error is reduced by 0.94. While in variant ViT, our approach achieves better performance with significantly fewer parameters. These results demonstrate the significance of selectively eliminating irrelevant tokens in discrete event data to enhance pose estimation.

**Table 6: Ablation results of using our adaptive tokens dropout module. X~Y indicates that there are X input patches and Y remaining tokens.**

| Input | A_patches | FLOPs | heads | depth | max_token | mlp_dim | MPJPE |
|---|---|---|---|---|---|---|---|
| 400 | 340 | 13.13G | 12 | 12 | (400~400) | 512 | 103.11 |
| 400 | 340 | 5.66G | 12 | 12 | (400~32) | 512 | 102.17 |
| 400 | 340 | 3.25G | 6 | 6 | (400~400) | 512 | 99.69 |
| 400 | 340 | 1.85G | 6 | 6 | (400~32) | 512 | 98.54 |

## 6 CONCLUSION

This paper presents Adaptive Vision Transformer for event-based human pose estimation. It demonstrates scalability, flexibility, and transfer ability for the pose estimation tasks, which have been well justified through extensive experiments on EventMM HPE dataset, which is first proposed for its high frame rate annotations (240Hz). Adaptive Vision Transformer for human pose estimation model with adaptive patches sampling and adaptive tokens dropout obtains the best 99.34 MPJPE on the EventMM HPE test set. We hope this work could provide useful insights to the community and inspire further study on exploring the potential of plain vision transformers in more computer vision tasks.

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
