# OpenReview forum: "Adaptive Vision Transformer for Event-Based Human Pose Estimation"
_acmmm.org/ACMMM/2024/Conference — MM2024 Poster_

### Official Review · Reviewer_6g7U · 2024-05-24

**Rating:** 4
**Confidence:** 3

**Summary:**

The paper discusses the challenges faced by traditional RGB-based human pose estimation methods in scenarios with rapid movements and challenging lighting conditions, and introduces event-based cameras as a solution due to their high temporal resolution, dynamic range, and low energy consumption. The paper proposes an adaptive vision transformer framework for event-based human pose estimation, utilizing adaptive patch sampling and token reduction strategies to efficiently extract spatial-temporal features from event data. A large-scale event-based human pose estimation dataset named EventMM HPE is introduced, providing synchronized event data and RGB images with a high annotation frame rate. The contributions of the paper include the novel adaptive vision transformer architecture, the dataset construction, and experimental results showing the outperformance of state-of-the-art methods.

**Strengths:**

1. Propose a large-scale dataset with accurate pose annotation and fast movement.
2. Propose a simple baseline based on a vision Transformer, including two novel components to sample and drop tokens for efficiency.
3. The experiments demonstrated the effectiveness of the proposed two components in the vision Transformer.

**Limitations:**

## My concern mainly lies in the insufficient evaluation.
1. For dataset advantages, in addition to its quantity, the accuracy of the pose annotation is also mentioned. However, there is no comparison of the accuracy of the annotations between RGB-D and Vicon in the experiment section.
2. Although the FLOPS are reported, the run time is not reported, which is more important to know for real-time applications.
3. The baseline methods (PointNet[15], DGCNN [21], and DHP19 [2]) seem not to be the SOTA methods as stated in 699. As for point cloud-based methods, there are many more advanced methods than PointNet[15] and DGCNN [21].
4. For experiments, I would also like to see a comparison between SOTA image-based pose estimation and event-based pose estimation, to demonstrate the advantages of using an event camera for pose estimation.
## Questions
1. What are the advantages of having a very high framerate for pose estimation tasks?
2. L663: what is the difference between n+1 and N+1?
3. L645~L669: Is this part related to "Adaptive tokens dropout"? Or they are just normal illustrations for a standard component of Transformer? If so, they are not supposed to be here since they are too standard and are not your contributions.
## Other comments
1. "Impact of Transformer Parameters" may not be an ablation study. This simple control experiment seems less meaningful in supporting the contributions of this paper.

**Suitability:**

2

---

### Official Review · Reviewer_eZWN · 2024-05-24

**Rating:** 4
**Confidence:** 3

**Summary:**

This paper proposed a high-frame rate labeled event-based human pose estimation dataset named Event Multi Movement HPE (EventMM HPE). It consists of records from synchronized event camera, high frame rate camera and Vicon motion capture system, with each sequence recording multiple action combinations and high frame rate (240Hz) annotations.  This paper also proposed an event-based human pose estimation model, which utilizes adaptive patches to efficiently achieves good performance for the sparse and reduced input data from DVS.

**Strengths:**

1.This paper propose a novel adaptive vision transformer architecture for human pose estimation, allowing us to effectively and efficiently extract spatial-temporal features from events.
2.This paper construct a large-scale event-based dataset for human pose estimation. The dataset provides a wide diversity in action and offers high annotation frame rate.

**Limitations:**

1. There is a lack of introduction to relevant datasets, such as dvsgesture128.
2. The Adaptive Vision Transformer section only includes adaptive patch sampling and adaptive token dropout. The method section lacks an introduction to the backbone network and other parts of the network.

**Suitability:**

2

---

### Official Review · Reviewer_QLWf · 2024-05-25

**Rating:** 4
**Confidence:** 3

**Summary:**

In this work, the author introduces a dataset as well as an event based human pose estimation method based on event cameras. This work modifies the vanilla vision transformer (ViT) to another transformer method namely Adaptive Transformers that uses a different patch sampling compared to the former.

**Strengths:**

Technically the paper seems to be sound. The overview diagram of the paper makes for the easy understanding of the pipeline. The paper in general is well structured.

**Limitations:**

1. The paper does not seem to compare with the ideas of token reorganisation.
2. How does vanilla patch dropout compare to the adaptive dropout.
3. Limited comparisons on the baselines.
4. How does the complexity change while using adaptive dropout.

**Suitability:**

3

---

### Meta-Review · Area_Chair_1qoe · 2024-07-05

**Recommendation:** Accept (Poster)
**Confidence:** 4

**Metareview:**

This paper addresses the problem of Event-based human pose estimation.
The authors wrote a solid rebuttal with additional experiments and significant clarifications that in my opinion addressed all reviewers concerns. Given that the final rating of one of the reviewers is not justified at all and the large-scale dataset contribution compensates for the moderate novelty of the methodological contribution, I recommend to accept this paper.